# External Apical Root Resorption in Orthodontic Patients Who Practice Combat Sports: A Case-Control Observational Pilot Study

**DOI:** 10.3390/medicina58101342

**Published:** 2022-09-24

**Authors:** Alfonso Enrique Acevedo-Mascarúa, Rafael Torres-Rosas, Yobana Pérez-Cervera, Daniel Pérez-Cruz, Lizbeth Zulema Ku-Valenzuela, Ana Lilia Gijón-Soriano, Liliana Argueta-Figueroa

**Affiliations:** 1División de Posgrado, Facultad de Odontología, Universidad Autónoma “Benito Juárez” de Oaxaca, Av. Universidad S/N, Ex-Hacienda 5 Señores, Oaxaca de Juárez 68120, Mexico; 2Laboratorio de Inmunología, Centro de Estudios en Ciencias de la Salud y la Enfermedad, Facultad de Odontología, Universidad Autónoma “Benito Juárez” de Oaxaca, Av. Universidad S/N, Ex-Hacienda 5 Señores, Oaxaca de Juárez 68120, Mexico; 3Consejo Nacional de Ciencia y Tecnología, Facultad de Odontología, Universidad Autónoma “Benito Juárez” de Oaxaca, Av. Universidad S/N, Ex-Hacienda 5 Señores, Oaxaca de Juárez 68120, Mexico

**Keywords:** dentistry, orthodontic, combat sports, root resorption

## Abstract

Orthodontic treatment could lead to undesirable effects such as external apical root resorption (EARR). Moreover, trauma to both the face and teeth can predispose to EARR. On the other hand, the practice of combat sports results in increased maxillofacial injuries. Consequently, our objective was to determine if there is a statistically significant difference in the EARR of the patients undergoing fixed orthodontic treatment who practice combat sports and controls. Our null hypothesis was that there is no difference in the EARR between patients undergoing orthodontic treatment who practice combat sports and the patients under the same treatment that do not practice combat sports. An observational, descriptive, and prospective case-control pilot study was designed. The exposed group consisted of patients that practice combat sports. Whereas the control group was conformed of patients that do not practice combat sports without a previous history of facial trauma and without face trauma during the orthodontic treatment. EARR of the maxillary and mandibular anterior teeth was measured using cone-beam computed tomography (CBCT). The CBCT scans were obtained from all patients prior to the beginning of the orthodontic treatment and 1 year later. At the end of the follow-up for the maxillary right central and lateral incisors of the exposed group, the EARR was significantly higher than the homologous teeth of the control group (*p* < 0.05). As a consequence, the patients treated orthodontically who practice combat sports could be more susceptible to EARR.

## 1. Introduction

Orthodontic treatment could lead to undesirable effects, such as root resorption, tooth loss, dental fractures, demineralization, enamel wear, root resorption, early closure of the apex, bone resorption, traumatized soft tissues, temporomandibular dysfunction, and condylar resorption [1]. External apical root resorption (EARR) commonly concerns orthodontists because it diminishes root length and volume, alters the center of resistance, and compromises support and anchorage [2].

EARR is a complication that occurs as a consequence of orthodontic treatment. However, the risk factors for EARR have not been completely clarified [3,4,5]. Some of the risk factors suggested related to orthodontic treatment are inadequate biomechanics, the prolonged duration of treatment, the intensity of the forces, as well as range and type of the movements [6,7,8,9]. On the other hand, the risk factors suggested related to the patients are age, personal habits, periodontal disease, occlusal relationship, dental size and morphology, periapical infection, and dental trauma [10,11,12,13,14,15,16,17,18,19]. However, several publications in dentistry have divergent results despite presenting similar characteristics or measuring different outcomes to determine the effect of treatment [20,21], and the systematic reviews of the EARR in orthodontic patients reported risk of bias and low to moderate quality in the included studies [22,23,24].

The frequency of dental traumatism In patients attending dentistry consultation is high, with a prevalence of 15.2% on permanent dentition and 22.7% on primary dentition [25]. Those traumatisms are related mainly to falls (up to 64%), followed by sports practice (up to 40.4%) and car accidents (up to 7.8%) [26]. Nevertheless, the severe trauma of the supporting tissue and teeth can predispose to EARR in patients without orthodontic treatment [27].

Most traumatic dental injuries in primary and permanent dentitions involve the anterior maxillary teeth, especially the central and lateral incisors. Traumatic dental injuries usually affect a single tooth, but sports practice may result in multiple teeth injuries [28]. Nevertheless, practicing combat sports results in an increase in both face and tooth injuries [29]. The risk for dental traumatism related to sports practice is classified into two categories. The high-risk sports include inline-skating, skate-boarding, martial sports, lacrosse, rugby, American football, ice hockey, mountain biking, and skating; and medium-risk sports, which include basketball, team handball, squash, gymnastics, water polo, and soccer [30]. Based on this background, the current study aimed to determine if there is a statistically significant difference in the EARR of the patients undergoing fixed orthodontic treatment who practice combat sports and controls. Then, our null hypothesis was that there is no difference in the EARR between patients undergoing orthodontic treatment who practice combat sports and the patients under the same treatment that do not practice combat sports.

## 2. Materials and Methods

This research was conducted at the Orthodontics Department of the Postgraduate’s Clinic of Dentistry at the “Universidad Autónoma Benito Juárez de Oaxaca”. The type of design was an observational, descriptive, and prospective case-control study. Ethical approval (ref. no: 19POC001FO) was obtained from the local Human Research Ethics Committee (Comité de ética en investigación de la Facultad de Odontología, Universidad Autónoma “Benito Juárez” de Oaxaca). The inclusion criteria were patients with molar class I (without severe crowding) under fixed orthodontic treatment with the Edge-Wise technique (January to December 2019), aged between 18 and 25 years old, without gender restrictions. The exclusion criteria were female patients under contraceptive treatment, patients with parafunctional habits, endodontic treatment, the presence of restoration at the incisal edge, previous or incomplete orthodontic treatment, as well as systemic, degenerative, and metabolic disorders or genetic syndromes. 

Exposed group: Patients that practice combat sports (boxing or martial arts).

Control group: Patients that do not practice combat sports without a previous history of facial trauma or patients without facial trauma during the orthodontic treatment.

The participants of the Exposed group were selected in sequence as they came for initial orthodontic consultation if they met the inclusion criteria. On the other hand, the participants of the Control group were selected randomly from the total patients that met the inclusion criteria.

The sample size was calculated using statistical power analysis [31], which, in a hypothesis test, is the probability that the test will detect an effect that actually exists, considering that 1.0 mm of EARR is clinically meaningful [32]. The minimum sample size calculated was 28 participants assuming power = 80%, alpha = 0.05, a standard deviation = unknowing, and mean difference = 1.0. However, the present research is only a pilot study, and due to that, the sample calculation was determined as proposed by Whitehead et al. [33] following the non-central t-distribution approach resulting in a sample size of 7 participants per group.

The evaluation of EARR was established through the measurements of cone-beam computed tomography (CBCT). The CBCT scans were obtained from all of the patients prior to the beginning of the orthodontic treatment (T1) and 1 year later (T2). The scans were carried out in the same imaging center (“Imagen Dentofacial Digital”) using Promax^®^ 3D Mid (PlanMeca, U.S.A). The scan parameters were a pixel size 127 um, active surface 15 × 15 cm, 90 kVp, and 5.2 mA. The CBCT scans were assessed by the same researcher (DP-C) to determine the EARR using the Planmeca Romexis Viewer Software (Romexis^®^ 5.2.1.R). The intraclass correlation coefficient (ICC) was calculated to perform the intra-rater test with repeated measures of the teeth studied. The ICC was performed using the Irr package of the R software (R Development Core Team, 2011. Version 4.2.0). 

Sagittal cuts of the teeth in a projection with a mediolateral orientation (at the longest vestibulopalatine width at the cementoenamel junction) were selected. We measured the distance from the dentin–enamel junction at the level of the incisal edge to the root apex. The EARR was calculated by assessing the difference in the tooth length between T1 and T2 (T1–T2) in millimeters, as shown in Figure 1. Both the researcher who performed the measurements and the one who analyzed the data were blinded, unaware of the study groups.

The normal distribution and variances of the data were tested using the Shapiro–Wilks test and the Bartlett or Levene tests, respectively. The results were expressed as the mean and standard deviation of the difference between the T1 and T2 measurements for both groups. A nonpaired t-test, Mann–Whitney U test, or Yuen t-Test was performed according to the distribution and variances of the data [34]. The analysis and graphs were performed using dplyr, nortest, ggstatsplot, and WRS2 packages of the R software (R Development Core Team, 2011. Version 4.2.0) and JASP (JASP Team, 2022. Version 0.16.3) software.

## 3. Results

### Patients

Five hundred and ninety patients were treated at the Orthodontics Department of the Postgraduate Clinic of Dentistry from January to December 2019. The sample consisted of 14 patients that met inclusion criteria: seven patients that practiced combat sports and seven patients with the same age range and diagnosis as controls. The mean age of the participants was 22 ± 2 years for the study group and 20 ± 0.97 years for the control group (*p* > 0.05).

The ICC results had good reliability (0.87). Meanwhile, according to the Shapiro–Wilks and Levene test results, the data did not follow a normal distribution but showed equal variances. The EARR (mean ± standard deviation) and statistical analysis are shown in Table 1 and Table 2.

The EARR of the maxillary teeth was statistically significant higher in the exposed group (Control = 1.251 ± 1.10; Exposed = 2.171 ± 1.82; *p* < 0.05). However, the difference between the two groups was not statistically significant in the mandibular teeth (Control = 1.93 ± 1.65; Exposed = 2.25 ± 3.12; *p* > 0.05; Table 1 and Figure 2).

The EARR in the right maxillary central and lateral incisors (0.84 ± 0.74; 0.54 ± 0.35, respectively) of the exposed group was significantly higher than in homologous teeth of the control group (*p* < 0.05). However, the difference between the two groups was not statistically significant in the rest of de evaluated teeth (*p* > 0.05; Table 2, Figure 3 and Figure 4).

## 4. Discussion

EARR is common in patients treated orthodontically; however, there is no specific test to evaluate the prognosis for the patients at risk for EARR. Currently, the follow-up for early diagnosis of EARR includes screening for a history of trauma [35]. Nonetheless, to the best of our knowledge, there is insufficient evidence that supports the relationship between EARR in patients treated with fixed orthodontics and the antecedents of dental trauma [22]. 

In this study, the EARR in the maxillary teeth was statistically significantly higher in the exposed group, which agrees with Brin et al. [36], who evaluated the EARR of traumatized permanent maxillary incisors related to orthodontic treatment with a 6-month follow-up after completion of the retention period. Their results suggest that combining dental trauma with orthodontic treatment causes the maxillary teeth to be more susceptible to root resorption. 

Dental trauma may increase the risk of EARR in patients treated orthodontically, which is exemplified by the fact that at the moment of impact, a considerable amount of energy is expended in driving the tooth into its dental alveolus. These traumatic forces may compress the periodontal ligament and damage the alveolar wall, diving into the loss of tissue [37]. In addition, mechanical forces promote osteoclast differentiation and activity that induces dental root resorptions [38,39]. 

In this study, the EARR in the right maxillary central and lateral incisors was statistically significantly higher in the exposed group. However, the difference between the two groups was not statistically significant in the rest of the evaluated teeth, which may be explained by the fact that despite the combat sports practiced, there was a high risk of dental traumatism. In this study, there was not enough information about the intensity, location, and type of trauma to the face and teeth during the practice. Furthermore, the sample was too small. Future research should correlate the classification and severity of trauma to the face and teeth with the EARR in patients undergoing orthodontic treatment.

Hevoca et al. [40] investigated pre-injury factors, the causes of dental injuries, and healing complications after traumatic injuries to permanent teeth. The research team reported a prevalence of EARR on intruded teeth (33.3%), luxated teeth (11.6%) avulsed, and replanted teeth (26.5%). However, teeth with isolated crown fractures did not present EARR. On the other hand, in teeth with root fractures, EARR was founded in 36.7%. These findings suggest that the presence of EARR may be related to the damage to both dental alveolus and root tissues. Unfortunately, there exist more than 50 traumatic dental injury classification systems. Most of them are related to treatment methods and the later complications of dental injuries [41]. Moreover, there is a lack of evidence of the EARR as a complication of facial trauma, which correlates with our findings that the difference between the two groups was not statistically significant in most evaluated teeth. 

On the other hand, in this study, it was found that the distribution of the EARR data did not follow a normal distribution, which coincides with the results of other studies in which the EARR was evaluated, taking into account other variables such as the type of orthodontic technique (self-ligated vs. conventional) [42,43] or teeth vitality (teeth with or without previous endodontic treatment) [44,45]. Due to this type of distribution, the data must be analyzed cautiously, using the appropriate statistical test, as well as not drawing rushed conclusions. In general, the fact that there are statistically significant differences between the means of two groups does not necessarily imply that there is clinical significance.

The strengths of this pilot study lie in the variables that were homogenized to avoid bias, such as the fixed orthodontic system used, the orthodontic technique, the CBTC device, the conditions for taking the CBTC, the time between each EARR measurement, the age range of participants, and exposure. To the best of our knowledge, there are no studies with an adequate design that support that a greater degree of EARR is produced in patients with fixed orthodontics with trauma since, until now, most of the evidence is empirical, anecdotal, and comes from case reports. On the other hand, the limitations of this study are that the characteristics of the facial trauma (number, location, type, and intensity of trauma) of the exposed group during the orthodontic treatment were not collected. Furthermore, the type of combat sport (boxing, karate, kickboxing, judo, taekwondo) may influence the probability, localization, and intensity of trauma [46,47]. Consequently, the type of combat sport may relate to the severity of EARR and the affected teeth. Nevertheless, the results in patients who practice combat sports may not be extrapolated to dental trauma by car accidents, falls, and violence. Moreover, other EARR miscellaneous factors should be addressed, such as oral habits and biological factors.

Despite the fact that this is a pilot study, another limitation is that small samples can overestimate the size of the effect due to the random error attributable to an insufficient sample, where the results could be scattered around the real effect [48,49]. Further research should be randomized clinical trials with an adequate sample size, confining the type of combat sport, examining the facial trauma characteristics, and evaluating the mechanical and biological factors of EARR.

## 5. Conclusions

The maxillary right central and lateral incisors of combat sports patients exhibited statistically significantly higher resorption than controls. However, the clinical relevance is unclear. This susceptibility could be related to the severity of the dental trauma. More studies with large samples are required to generate solid conclusions.

## Figures and Tables

**Figure 1 medicina-58-01342-f001:**
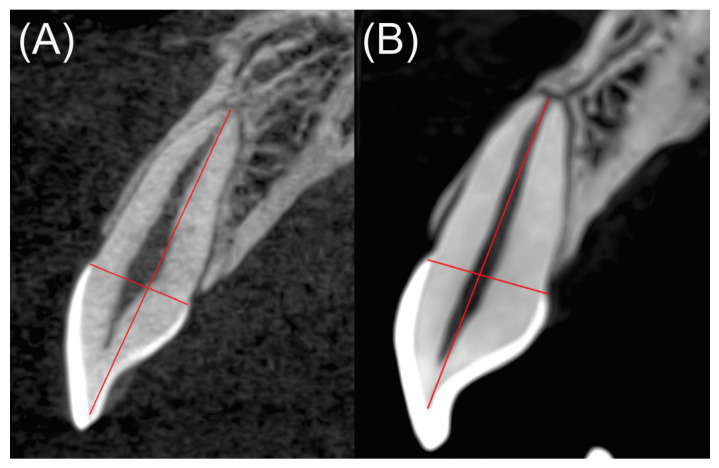
Sagittal cut of a tooth in a projection with a mediolateral orientation. The root length was measured by drawing a line from the incisal edge (at the dentin-enamel junction) to the apex, (**A**) tooth without EARR, (**B**) tooth with EARR.

**Figure 2 medicina-58-01342-f002:**
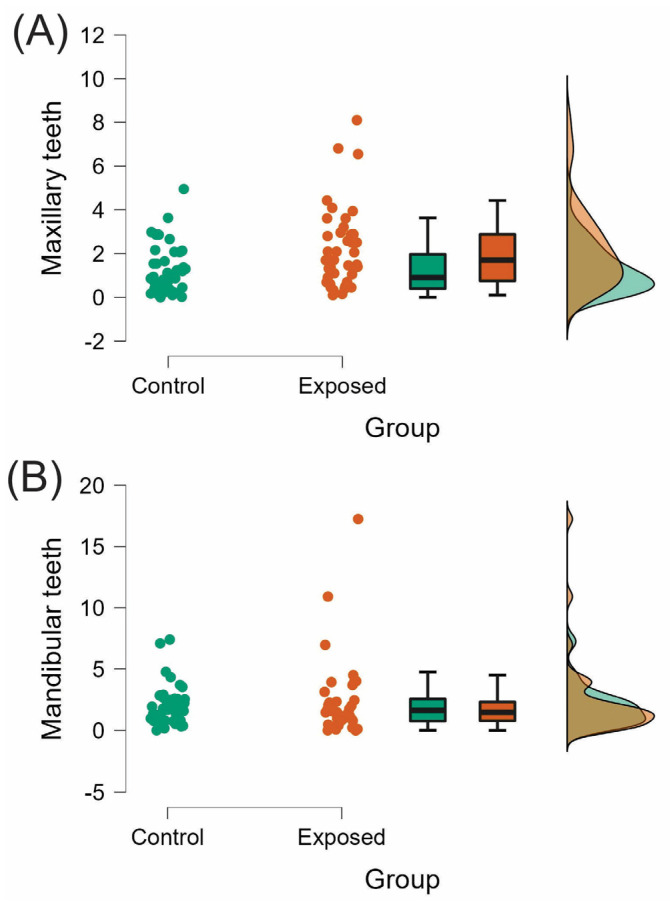
Raincloud plots of the EARR in maxillary anterior teeth. (**A**) Maxillary teeth and (**B**) Mandibular teeth. EARR expressed as mean difference between T1 and T2 measurements (mm) and SD.

**Figure 3 medicina-58-01342-f003:**
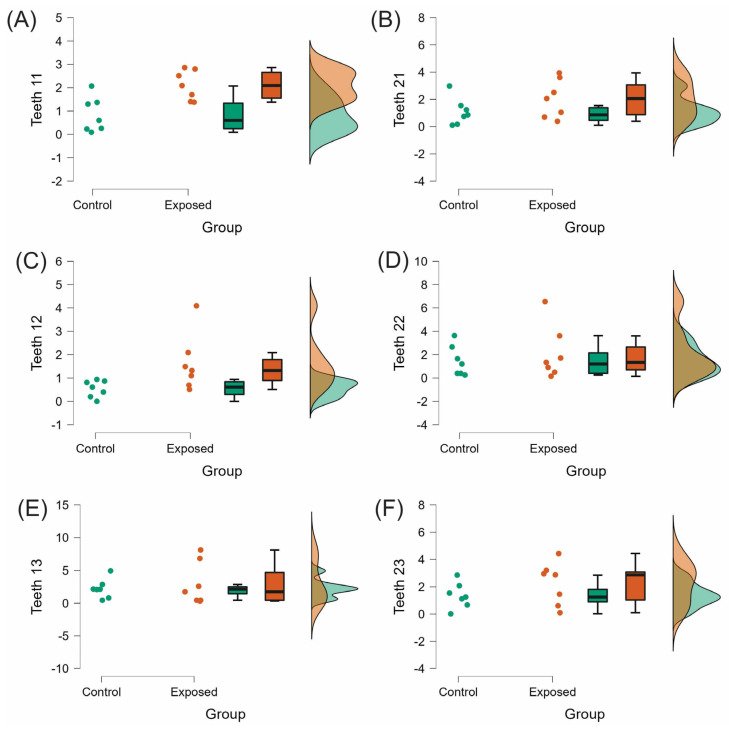
Raincloud plots of the EARR in maxillary anterior teeth. (**A**,**B**) central incisors, (**C**,**D**) lateral incisors, and (**E**,**F**) canines. EARR expressed as mean difference between T1 and T2 measurements (mm) and SD.

**Figure 4 medicina-58-01342-f004:**
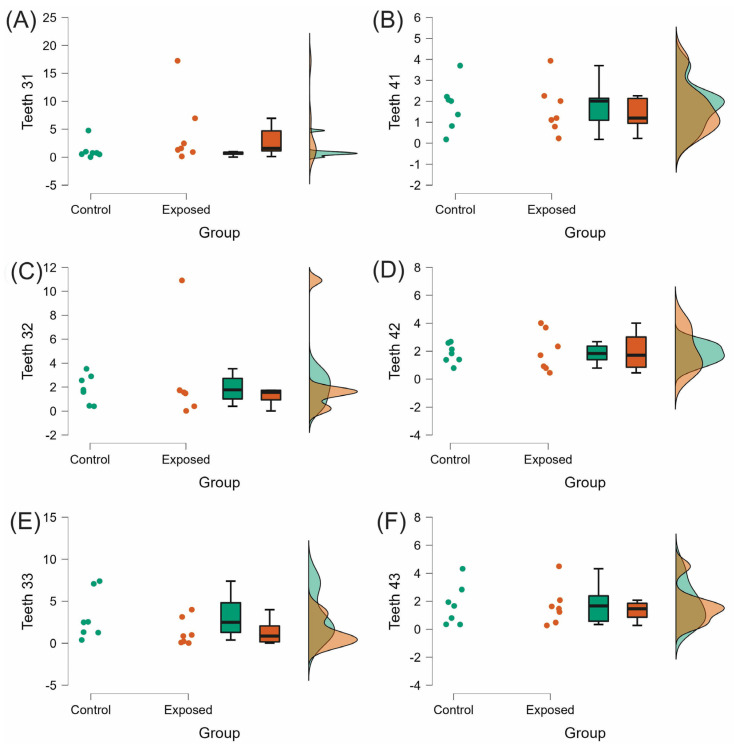
Raincloud plots of the EARR in mandibular anterior teeth. (**A**,**B**) central incisors, (**C**,**D**) lateral incisors, and (**E**,**F**) canines. EARR expressed as mean difference between T1 and T2 measurements (mm) and SD.

**Table 1 medicina-58-01342-t001:** Results of the comparison between the EARR of the exposed and control participants (Maxillary vs. Mandibular teeth).

	Maxillary teeth		Mandibular teeth	
	**Control**	**Exposed**	**Control**	**Exposed**
Mean	1.251	2.171	1.939	2.258
SD	1.100	1.827	1.653	3.120
Minimum	0.000	0.100	0.020	0.010
Maximum	4.950	8.100	7.400	17.240
			W	*p*
Maxillary teeth			589.000	0.004 *
Mandibular teeth			938.500	0.695

U de Mann-Whitney test. Standard deviation (SD). * Statistically significant difference.

**Table 2 medicina-58-01342-t002:** Results of the comparison between the EARR of the exposed and control participants.

**Teeth**	**11**	**12**	**13**
Control (mean ± SD)	0.84 ± 0.74	0.54 ± 0.35	2.20 ± 1.47
Exposed (mean ± SD)	2.10 ± 0.63	1.61 ± 1.21	2.92 ± 3.22
*p*-value	0.004 *	0.026 *	0.797
**Teeth**	**21**	**22**	**23**
Control (mean ± SD)	1.09 ± 0.98	1.45 ± 1.28	1.36 ± 0.92
Exposed (mean ± SD)	2.03 ± 1.40	2.11 ± 2.25	2.23 ± 1.55
*p*-value	0.259	0.701	0.259
**Teeth**	**31**	**32**	**33**
Control (mean ± SD)	1.18 ± 1.60	1.88 ± 1.19	3.21 ± 2.85
Exposed (mean ± SD)	4.36 ± 6.09	2.54 ± 3.75	1.33 ± 1.59
*p*-value	0.097	0.383	0.128
**Teeth**	**41**	**42**	**43**
Control (mean ± SD)	1.76 ± 1.13	1.83 ± 0.69	1.75 ± 1.45
Exposed (mean ± SD)	1.64 ± 1.22	1.98 ± 1.42	1.66 ± 1.40
*p*-value	0.902	0.949	0.902

Mann–Whitney U test. Standard deviation (SD). * Statistically significant difference.

## Data Availability

The data that support the findings of this study are available from the corresponding author upon reasonable request.

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
