# Peer review of "External Apical Root Resorption in Orthodontic Patients Who Practice Combat Sports: A Case-Control Observational Pilot Study"

_medicina, 2022, doi:10.3390/medicina58101342_

Round 1
Reviewer 1 Report
I wonder if only the maxillary right central and lateral incisors were investigated, what about the maxillary left central and lateral incisors were they not involved in the study ? Were they not affected by the combat sports ? They were ignored, why ? Please explain.
The authors mentioned that the study sample was too small, with only 7 patients in each group (study & control).
Author Response
Thank you for the peer-review and the suggestions to improve the present manuscript. The answers of the reviewer´s comments and subsequent changes are shown below. The revised sections have been highlighted with blue in the main text. We hope that the changes are adequate to respond each comment.
1.- I wonder if only the maxillary right central and lateral incisors were investigated, what about the maxillary left central and lateral incisors were they not involved in the study ? Were they not affected by the combat sports ? They were ignored, why ? Please explain.
We included all the anterior teeth in the study as showed in the Table 2. However, only the EARR in the right maxillary central and lateral incisors was statistically significant higher in the exposed group. The EARR on the other teeth may not be affected due to facial trauma characteristics, but those data were not collected, this study limitations has been added in the discussion section. Thank you.
2.- The authors mentioned that the study sample was too small, with only 7 patients in each group (study & control).
The sample size was calculated using statistical power analysis[1], which in a hypothesis test is the probability that the test will detect an effect that actually exists, considering that 1.0 mm of EARR is clinically meaningful[2]. The minimum sample size calculated was 28 participants assuming power=80%, alpha=0.05, a standard deviation=unknowing, and mean difference=1.0. However, the present research only is a pilot study, due to that, the sample calculation was determined as proposed Whitehead, et al.[3] following the non-central t-distribution approach resulting in a sample size of 7 participants per group. Due to that, a study with adequate sample for a main trial are required to generate solid conclusions. Thank you.
- Schoenfeld DA. Statistical considerations for a parallel trial where the outcome is a measurement MGH Biostatistics Center: Massachusetts General Hospital Mallinckrodt General Clinical Research Center; [Available from: http://hedwig.mgh.harvard.edu/sample_size/js/js_parallel_quant.html.
- Mohandesan H, Ravanmehr H, Valaei N. A radiographic analysis of external apical root resorption of maxillary incisors during active orthodontic treatment. European Journal of Orthodontics. 2007;29(2):134-9.
- Whitehead AL, Julious SA, Cooper CL, Campbell MJ. Estimating the sample size for a pilot randomised trial to minimise the overall trial sample size for the external pilot and main trial for a continuous outcome variable. Statistical Methods in Medical Research. 2016;25(3):1057-73.
Reviewer 2 Report
Dear authors, I would like to congratulate for the present study on external apical root resorption.
The title and keywords appear correct. The abstract subheading should be removed according to the journal guidelines, and the Results sections should be improved and specific numeric values should be presented.
The manuscript body references should be placed inside square brakets [ ].
If the authors find appropriated I suggest the exposition of a null hypothesis after the aim sentence of the abstract.
How were the patients selected and what was the sampling method. All the patients were selected? In sequence? Random?
Did the authors conducted a sample size calculation?
Did the authors conducted intra-rater tests to test the observer reliability?
Did the authors noted any source of bias, and if yes which and how was it managed?
The only CBCT image is Figure 1. May the authors show some Figure of apical resorptions to be clearer what the authors assessed?
The study strength, limitations and further study research should be debated in the Discussion
The reference list is not according to the journal guidelines.
Author Response
Thank you for the peer-review and the suggestions to improve the present manuscript. The answers of the reviewer´s comments and subsequent changes are shown below. The revised sections have been highlighted with blue in the main text. We hope that the changes are adequate to respond each comment.
1.- The title and keywords appear correct. The abstract subheading should be removed according to the journal guidelines, and the Results sections should be improved and specific numeric values should be presented.
We removed the abstract subheading and we presented the main numeric values in the Results section, thank you.
2.- The manuscript body references should be placed inside square brakets [ ].
Done, thank you.
3.- If the authors find appropriated I suggest the exposition of a null hypothesis after the aim sentence of the abstract.
We added the null hypothesis in the abstract and at the end of the introduction section. Thank you.
4.- How were the patients selected and what was the sampling method. All the patients were selected? In sequence? Random?
The participants of the Exposed group were selected in sequence as they came for initial orthodontic consultation if they meet the inclusion criteria. On the other hand, the participants of the Control group were selected randomly from the total of the patients that meet the inclusion criteria. We added this information in the methods section. Thank you.
5.- Did the authors conducted a sample size calculation?
The sample size was calculated using statistical power analysis[1], which in a hypothesis test is the probability that the test will detect an effect that actually exists, considering that 1.0 mm of EARR is clinically meaningful[2]. The minimum sample size calculated was 28 participants assuming power=80%, alpha=0.05, a standard deviation=unknowing, and mean difference=1.0. However, the present research only is a pilot study, due to that, the sample calculation was determined as proposed Whitehead, et al.[3] following the non-central t-distribution approach resulting in a sample size of 7 participants per group.
We added the information in the methods section, thank you.
6.- Did the authors conducted intra-rater tests to test the observer reliability?
We calculate the Intraclass Correlation Coefficient in R resulting in a good reliability (0.87) according to Koo & Li. We added the related information in the methods and results sections, thanks you.
7.- Did the authors noted any source of bias, and if yes which and how was it managed?
The variables that were homogenized to avoid bias were: the fixed orthodontic system used, the orthodontic technique, the CBTC device, the conditions for taking the CBTC, the time between each EARR measurement, the age range of participants, and exposure. Also, the researcher who performed the measurements and the one who analyzed the data were blinded, unaware of the study groups. The above is described in methods and discussion sections.
8.- The only CBCT image is Figure 1. May the authors show some Figure of apical resorptions to be clearer what the authors assessed?
We added the figure 2B of apical resorption, thank you.
9.- The study strength, limitations and further study research should be debated in the Discussion
We added the following text:
The strengths of this pilot study lie in the variables that were homogenized to avoid bias such as the fixed orthodontic system used, the orthodontic technique, the CBTC device, the conditions for taking the CBTC, the time between each EARR measurement, the age range of participants, and exposure. To the best of our knowledge, there are no studies with an adequate design that support that a greater degree of EARR is produced in patients with fixed orthodontics with trauma since until now, most of the evidence is empirical, anecdotal, and comes from case reports. On the other hand, the limitations of this study are that despite the facial trauma characteristics (number, location, type, and intensity of trauma) of the exposed group during the orthodontic treatment, which were not collected. Also, the type of combat sport (boxing, karate, kickboxing, judo, taekwondo) may influence the probability, localization, and intensity of trauma[4, 5]. Consequently, the type of combat sport may relate to the severity of EARR and the affected teeth. Nevertheless, the results in patients who practice combat sports may not be extrapolated to dental trauma by car accidents, falls, and violence. Also, other EARR miscellaneous factors should be addressed, like oral habits, and biological factors.
Despite that this is an pilot study, another limitation is that small samples can overestimate the size of the effect due to the random error attributable to an insufficient sample, where the results could be scattered around the real effect[6, 7]. Further research should be randomized clinical trials with an adequate sample size, confining the type of combat sport, examining the facial trauma characteristics, and evaluating mechanicals and biological factors of EARR.
10.- The reference list is not according to the journal guidelines.
We modified the reference list according to the journal guidelines, thank you.
- Schoenfeld DA. Statistical considerations for a parallel trial where the outcome is a measurement MGH Biostatistics Center: Massachusetts General Hospital Mallinckrodt General Clinical Research Center; [Available from: http://hedwig.mgh.harvard.edu/sample_size/js/js_parallel_quant.html.
- Mohandesan H, Ravanmehr H, Valaei N. A radiographic analysis of external apical root resorption of maxillary incisors during active orthodontic treatment. European Journal of Orthodontics. 2007;29(2):134-9.
- Whitehead AL, Julious SA, Cooper CL, Campbell MJ. Estimating the sample size for a pilot randomised trial to minimise the overall trial sample size for the external pilot and main trial for a continuous outcome variable. Statistical Methods in Medical Research. 2016;25(3):1057-73.
- Ha S, Kim MJ, Jeong HS, Lee I, Lee SY. Mechanisms of Sports Concussion in Taekwondo: A Systematic Video Analysis of Seven Cases. Int J Environ Res Public Health. 2022;19(16).
- Shirani G, Kalantar Motamedi MH, Ashuri A, Eshkevari PS. Prevalence and patterns of combat sport related maxillofacial injuries. J Emerg Trauma Shock. 2010;3(4):314-7.
- Manzo-Toledo A, Torres-Rosas R, Mendieta-Zerón H, Arriaga-Pizano L, Argueta-Figueroa L. Hydroxychloroquine in the treatment of COVID-19 disease: a systematic review and meta-analysis. Medical Journal of Indonesia. 2021;30(1):20–32-20–32.
- Deeks JJ, Higgins JP, Altman DG, Group CSM. Analysing data and undertaking meta‐analyses. Cochrane handbook for systematic reviews of interventions. 2019:241-84.
Round 2
Reviewer 2 Report
Dear authors, I have no more comments. Thank you